# Alleviation Syndrome of High-Cholesterol-Diet-Induced Hypercholesterolemia in Mice by Intervention with *Lactiplantibacillus plantarum* WLPL21 via Regulation of Cholesterol Metabolism and Transportation as Well as Gut Microbiota

**DOI:** 10.3390/nu15112600

**Published:** 2023-06-01

**Authors:** Kui Zhao, Liang Qiu, Yao He, Xueying Tao, Zhihong Zhang, Hua Wei

**Affiliations:** 1State Key Laboratory of Food Science and Technology, Nanchang University, Nanchang 330047, China; 2Centre for Translational Medicine, Jiangxi University of Traditional Chinese Medicine, Nanchang 330047, China

**Keywords:** *Lactiplantibacillus plantarum* WLPL21, hypercholesterolemia, cholesterol, SCFAs, gut microbiota

## Abstract

Probiotics are prospective for the prevention and treatment of cardiovascular diseases. Until now, systematic studies on the amelioration of hypercholesterolemia have been rare in terms of (cholesterol metabolism and transportation, reshaping of gut microbiota, as well as yielding SCFAs) intervention with lactic acid bacteria (LAB). In this study, strains of *Lactiplantibacillus plantarum*, WLPL21, WLPL72, and ZDY04, from fermented food and two combinations (*Enterococcus faecium* WEFA23 with *L. plantarum* WLPL21 and WLPL72) were compared for their effect on hypercholesterolemia. Comprehensively, with regard to the above aspects, *L. plantarum* WLPL21 showed the best mitigatory effect among all groups, which was revealed by decreasing total cholesterol (TC) and low-density lipoprotein cholesterol (LDL-C) levels, upregulated cholesterol metabolism (*Cyp27a1*, *Cyp7b1*, *Cyp7a1*, and *Cyp8b1*) levels in the liver, cholesterol transportation (*Abca1*, *Abcg5*, and *Abcg8*) in the ileum or liver, and downregulated *Npc1l1*. Moreover, it reshaped the constitution of gut microbiota; specifically, the ratio of *Firmicutes* to *Bacteroidetes* (*F*/*B*) was downregulated; the relative abundance of *Allobaculum*, *Blautia*, and *Lactobacillus* was upregulated by 7.48–14.82-fold; and that of *Lachnoclostridium* and *Desulfovibrio* was then downregulated by 69.95% and 60.66%, respectively. In conclusion, *L. plantarum* WLPL21 improved cholesterol metabolism and transportation, as well as the abundance of gut microbiota, for alleviating high-cholesterol-diet-induced hypercholesterolemia.

## 1. Introduction

Hypercholesterolemia, which has always occurred with high LDL cholesterol and even further formed ox-LDL—a leading cause of numerous chronic diseases, e.g., cardiovascular diseases (CVDs) [1], hypertension [2], diabetes [3], and kidney failure [4]—is recognized as a complex metabolic disorder owing to the accumulation of cholesterol in cells and plasma of blood [5]. The WHO reported that 17.9 million people died from CVDs in 2019, representing 32% of global deaths [6]. Plenty of factors (diet, weight, gender, age, exercise, and drinking) may affect the incidence and development of hypercholesterolemia [7,8,9,10,11]. The common treatment for ameliorating hypercholesterolemia includes drugs of statins, ezetimibe, proprotein convertase subtilisin/kexin type 9 (PCSK9), and peroxisome proliferatively activated receptor (PPARα) inhibitors, achieving a good effect; however, its side-effects (e.g., diarrhea, memory dysfunction, and diabetes risk) should not be neglected [12,13]. Therefore, exploring novel bioinoculants or probiotics to alleviate hypercholesterolemia is promising and should be a top priority for curbing CVDs.

Cholesterol is an essential structural component of the plasma membrane. Excessive cholesterol results in a series of pathological consequences, e.g., hypercholesterolemia [14]. It was reported that some enzymes (e.g., CYP7A1, ABCG5, and FXR) had a tight connection with cholesterol metabolite and transportation. Cholesterol 7α-hydroxylase (CYP7A1) in the classic pathway and sterol 27α-hydroxylase (CYP27A1) in the alternative pathway converted cholesterol into cholic acid and chenodeoxycholic acid [15]. ABCG5/8 transported cholesterol by activating the transcription of LXR [14], and FXR regulated cholesterol via the SR-B1/CYP7A1/FXR pathway to alleviate hypercholesterolemia in high-fat-diet-fed rats [16,17]. Therefore, monitoring changes of the above enzymes in transcription level should contribute to the disclosure of cholesterol metabolite, and the transportation, regulation, and further understanding of how hypercholesterolemia syndrome changes.

In the past decades, probiotics (i.e., *Lactobacillus*, *Enterococcus*, and *Bifidobacterium*) were approached for their function of improving chronic diseases (e.g., hypercholesterolemia and atherosclerosis). For instance, *E. faecium* strain 132 alleviated hypercholesterolemia by regulating cholesterol metabolite and positively improving the constitution of gut microbiota [18], and *E. faecium* GEFA01 exhibited a cholesterol-lowering effect through in vivo and in vitro models [19]. In addition, *L. plantarum* ATCC 14917 ameliorated atherosclerosis by regulating inflammation and oxidative stress [20]. *L. plantarum* HT121 had a great role in cholesterol metabolism and regulation of the gut microbiota to attenuate hypercholesterolemia in a high-cholesterol-diet-induced rat model [21].

Recent publications of LAB on attenuating hypercholesterolemia have increased gradually, and to our limited knowledge, systematic research for improving hypercholesterolemia syndrome is rare, namely, from the view of synthesis, decomposition, and transportation of cholesterol; the histopathological pattern of the liver and ileum; the inflammatory response; the constitution of gut microbiota; and the change in short-chain fatty acids (SCFAs) content. Until now, little has been known about the amelioration of hypercholesterolemia syndrome in the host by the combination of LAB strains.

In our previous studies, *L. plantarum* ZDY04 from fermented soybeans was demonstrated to regulate cholesterol transportation to alleviate atherosclerosis, and *E. faecium* WEFA23 from infants relieved hyperlipidemia via the cholesterol 7-alpha-hydroxylase gene by altering the composition of gut microbiota [22,23]. Until now, a systematic comparison of the combination of Lactobacillus and Enterococcus with a single strain for alleviating hypercholesterolemia has yet to be reported.

Based on the screening of two potential probiotics strains, namely, *L. plantarum* WLPL21 from Chinese fermented soybeans and *L. plantarum* WLPL72 from pickled pepper, in this study, we designed a comparison test in mice based on grouping: (1) a combination of those strains with *E. faecium* WEFA23; (2) the single strains above, using a high-cholesterol diet as a control and *L. plantarum* ZDY04 as a positive control. Therein, biochemical assays, histopathology, RT-qPCR, GC-MS, and bioinformation were approached to disclose the mechanism of probiotics alleviating hypercholesterolemia via regulating the change in cholesterol, SCFAs, and gut microbiota.

## 2. Materials and Methods

### 2.1. Bacterial Cultures

*E. faecium* WEFA23 from healthy infants was grown on brain heart infusion (BHI) agar (Oxoid, Basingstoke, UK) and incubated anaerobically at 37 °C for 12 h [24]. Strains of *L. plantarum* ZDY04, WLPL72, and WLPL21 from Chinese fermented food (pickled vegetables and pepper, soybeans) were cultured under anaerobic conditions in sterile deMan, Rogosa, and Sharpe agar (MRS broth; Beijing Solarbio Science & Technology Co., Ltd., Beijing, China) at 37 °C for 12 h.

### 2.2. Animals and Experimental Design

The animal protocol was approved by the Animal Ethics Committee of Nanchang University, Nanchang, Jiangxi Province, P. R. China (No. 0064257), and the diagram of the animal experiment scheme is shown in Figure 1. Conventional C57BL/6 mice (male; 19–21 g; aged 7–8 weeks) were purchased from SLAC Jingda Laboratory Animal company, Hunan, China. Animals were housed under standard conditions (12 h light/dark cycle at a temperature of 23 ± 1 °C and a humidity of 54 ± 2%) and used for experimentation after a week of acclimatization. Then, 70 male C57BL/6 mice were divided randomly into 7 groups with 10 mice per group. The 7 groups were treated as follows: ND (normal diet, 66% carbohydrates, 22% protein, 12% fat), HCD (high-cholesterol diet, including 16.9% lard, 1.3% cholesterol, and 0.5% sodium cholate), ZDY04 (HCD with *L. plantarum* ZDY04), WLPL21 (HCD with *L. plantarum* WLPL21), WLPL72 (HCD with *L. plantarum* WLPL72), W21E23 (HCD with a bacterial cocktail of *L. plantarum* WLPL21 and *E. faecium* WEFA23), and W72E23 (HCD with a bacterial cocktail of *L. plantarum* WLPL72 and *E. faecium* WEFA23). The mice in ND and HCD groups were gavaged with 0.2 mL of 0.01 M PBS, while mice in the groups of ZDY04, WLPL21, and WLPL72, as well W21E23 and W72E23, had an oral administration with 5 × 10^9^ cfu/mL (0.2 mL) of LAB and a cocktail mixture of *Lactobacilli* and *Enterococcus* (1:1) every day for 12 weeks, respectively.

### 2.3. Serum and Hepatic Lipids

The concentrations of serum and hepatic TC, triglycerides (TG), high-density lipoprotein cholesterol (HDL-C), LDL-C, aspartate aminotransferase (AST), and alanine aminotransferase (ALT) were measured using commercially available kits according to the corresponding commercial instructions (Nanjing Jiancheng Bioengineering Institute, Nanjing, China).

### 2.4. Fecal Total Bile Acid

Approximately 0.05 g of fresh fecal was collected and a 500 μL mixture of methanol and chloroform (2:1) was added to homogenate before heating in the water bath for 1 h at 45 °C. Then, the supernatant was isolated after being centrifuged at 2500 rpm for 10 min and its content of total bile acid was measured using commercially available kits (Nanjing Jiancheng Bioengineering Institute, Nanjing, China).

### 2.5. Histological Analysis

Liver and ileum tissue was dissected and fixed in 10% (*v*/*v*) paraformaldehyde phosphate buffer for 24 h and embedded in paraffin. Approximately 5 μm thick sections were prepared from each block and stained with hematoxylin and eosin, and serial 10 μm thick liver sections were cut and stained with oil red O.

### 2.6. Real-Time Quantitative Polymerase Chain Reaction (RT-qPCR)

Total RNA was isolated from the liver and ileum of mice using the Takara MiniBEST Universal RNA Extraction Kit according to the manufacturer’s protocol. RNA (1000 ng) was reverse-transcribed into cDNA using the Takara PrimeScriptTM RT reagent Kit with gDNA Erase. Quantitative PCR (qPCR) was performed on an ABI Prism 7900HT sequence detection system (Applied Biosystems, Darmstadt, Germany) with the following program: 30 s at 95 °C, followed by 40 cycles of 10 s at 95 °C and 1 min at 58–62 °C. All primer sequences were synthesized by Sangon Biotech Co., Ltd. (Shanghai, China) (Table 1), in which the housekeeping gene *Gapdh* was included for normalization. The relative quantification of the targeted gene expression was calculated by the 2^−ΔΔCt^ method.

### 2.7. SCFAs Analysis

SCFAs of cecum content were measured by GC-MS using an Agilent 7890-70000D instrument (Agilent Technologies, Santa Clara, CA, USA) with slight modifications. Briefly, 500 mg of cecum content was mixed with 2.5 mL of solvent (methanol: 0.2% HCl = 1:1) before ultrasonication at 250 W for 5 min and then centrifuged at 8000 rpm for 10 min. The supernatants were used for GC-MS analysis. One microliter of samples or standards was injected automatically at an inlet temperature of 240 °C and a split ratio of 50:1. The flow rate of helium was 20 cm/s at a temperature of 175 °C. The oven temperature program was as follows: initially set to 100 °C and held for 5 min, then increased to 150 °C at a rate of 10 °C/min, and held at 150 °C for 3 min. The ion source temperature was 220 °C. The SCFAs in the cecum were quantified by the standard curves of acetic acid, butyric acid, and propionic acid [25].

### 2.8. Analysis of Gut Microbiota by 16S rRNA Sequencing

The 16S rRNA gene amplicon sequencing was performed on the Illumina NovaSeq PE250 (APExBIO Technology LLC, Shanghai, China). The sequences of the primers targeting the V3/V4 hyper-variable region of the bacterial 16S rRNA genes were as follows: (341F, 5′-ACTCCTACGGGAGGCAGCA-3′. 805R, 5′-GGACTACHVGGGTWTCTAAT-3′). Sequence data analysis mainly used QIIME 2 (QIIME2-2020.2) and R software (4.0.2). QIIME 2 was used to calculate the alpha diversity index, plotting ASV-based ranked abundance curves and alpha dilute curves based on alpha diversity. Drawing a relatively abundant histogram (R GGPlot2 package), a group of Kruskal–Wallis inspection (multi-group) and Wilcoxon test (two groups) alpha diversity differential diagrams (R PHYLOSEQ Pack), and PCA and NMDS analysis (PHYLOSEQ/VGAN Pack) were finished by R. LDA Effect Size (LEfSe) analysis was performed with the help of the online analysis website: http://huttenhower.sph.harvard.edu/galaxy (accessed on 20 December 2021). 

### 2.9. Statistical Analysis

The data and the normal distribution of the data (*p* of Shapiro–Wilk normality greater than 0.05 was thought to conform to a normal distribution) were analyzed using GraphPad Prism 7 statistical software (Prism 7.0). The results were expressed as the mean ± standard deviation (SD). All the data that passed the normal distribution verification were analyzed via one-way analysis of variance, and Tukey’s multiple comparisons tests were used for comparisons between groups, while the others were analyzed via a non-parametric test. The *p*-values less than 0.05 were considered statistically significant.

## 3. Results

### 3.1. Probiotics Reduced Cholesterol Levels in Serum and Liver

Upon establishment of the hypercholesterolemia model of mice (Appendix A), the effect of three strains of *L. plantarum* and the combination of partial strains with *E. faecium* on decreasing levels of cholesterol was investigated. Although there was a non-significant difference in food or water intake, all five groups of probiotics demonstrated a significant loss of body weight (*p* < 0.0001 in week 12, Appendix A), as compared with the HCD group. However, indexes of the spleen and kidney showed no significant reduction by the intervention of probiotics (Appendix A).

Considering that blood lipids were critical factors for the evaluation of hypercholesterolemia, furthermore, to evaluate the capacity of lowering cholesterol level with the above probiotic strains, TC, TG, HDL-C, and LDL-C in the serum and liver were detected (Figure 2). It was observed that TC level in the serum and liver was significantly decreased in all probiotic treatments (*p* < 0.05), similar to the above result of l body weight loss. In contrast, either TG or HDL-C level in the serum and liver was non-significantly different among all groups (*p* > 0.05). As for LDL-C level, a significant reduction in the serum and liver was achieved only in the treatment with *L. plantarum* WLPL21.

As TBA reflected the capacity of cholesterol metabolism in the liver, we found that the supplementation of *L. plantarum* WLPL21, ZDY04, and WLPL72 significantly increased the level of fecal TBA (Appendix A, *p* < 0.05), indicating that probiotics could accelerate cholesterol metabolism in C57BL/6 mice.

### 3.2. Probiotics Alleviated Hepatic Injury Caused by HCD

The stacking of cholesterol in the serum and liver in the HCD group resulted in hepatic injury, as shown in Figure 3. HCD dramatically increased the liver index and its pathogenic histology behavior, and *L. plantarum* ZDY04 and WLPL21 treatment reversed the trend significantly (*p* < 0.05). Routine hepatic injury was reflected by activities of AST and ALT. As shown in Figure 3B, *L. plantarum* WLPL21 significantly decreased the level of ALT and AST (*p* < 0.01), and *L. plantarum* WLPL72 had the same effect on AST. Correspondingly, *L. plantarum* WLPL21 greatly reduced HCD-induced inflammatory cell infiltration and lipid droplets accumulation according to the HE staining pattern (Figure 3C). Moreover, from the view of hepatic steatosis, all five probiotic strains significantly reduced its score as compared with the HCD group, indicating that probiotics alleviated hepatic injury (Figure 3D).

Although a minor amelioration of stacking of cholesterol in the liver, as reflected by staining with Oil Red O, was observed for all probiotic treatments (Appendix A), the treatment of *L. plantarum* WLPL21 and ZDY04 greatly improved the histological pattern of the ileum (Appendix A).

### 3.3. Probiotics Regulated Cholesterol Metabolism in the Liver and Cholesterol Transportation in the Liver and Ileum

The above study led to the conclusion that the supplementation of probiotics greatly alleviated hepatic injury by the reduction in cholesterol accumulation. To explore its potential mechanism of alleviating the stacking of cholesterol in the liver and ileum, the metabolism and transportation of cholesterol were approached at the transcription level. Since *Cyp7a1*, *Cyp8b1*, *Cyp27a1*, and *Cyp7b1* were key metabolic genes in hepatic bile acid synthesis, hepatic samples of five groups after gavage with probiotics for 12 weeks were checked and compared for the expression of the above four genes. As shown in Figure 4, only the *L. plantarum* WLPL21 group significantly upregulated the expression of all genes related to cholesterol metabolism in the liver (Figure 4A, *p* < 0.05). On account of *Abca1*, *Abcg5*, *Abcg8*, and *Sr-b1* associated with cholesterol efflux and reverse transportation, only *L. plantarum* WLPL21 significantly (*p* < 0.01) upregulated the expression of the above genes except for *Sr-b1* (Figure 4B). Considering that *Fxr*, *Lxr*, and *Shp* were core-regulating genes in cholesterol metabolism, only *L. plantarum* WLPL21 upregulated the expression of *Lxr* and *Shp* and downregulated the expression of *Fxr* (*p* < 0.05) (Figure 4C). Since sterol-regulatory element binding proteins (SREBPs) had a great role in regulating cholesterol metabolism, our results showed that all probiotics downregulated its expression in Figure 4C. In addition, *Ppar-α* was a crucial gene to regulating the synthesis and metabolism of lipids, and all probiotics upregulated its expression in the liver (Figure 4C, *p* < 0.01). The ileum was also essential for cholesterol transportation; therefore, we validated whether our strains had a similar effect. As we estimated, all our strains significantly upregulated the expression of *Abcg5* and *Abcg8*, as well as downregulated the expression of *Npc1l1* (Figure 4D, *p* < 0.05).

### 3.4. Probiotics Modulated the Composition of Gut Microbiota and Cecum SCFA Level

Upon the above finding of the alleviation of hypercholesterolemia by probiotics (especially *L. plantarum* WLPL21), the SCFAs level and gut microbiota composition were approached to explore its potential mechanism. As shown in Figure 5, *L. plantarum* WLPL21, WLPL72, and a combination of *L. plantarum* WLPL21 and *E. faecium* WEFA23 significantly enhanced the level of organic acid (*p* < 0.05); specifically, *L. plantarum* WLPL21 increased the yield of acetic acid, propionate, and butyrate by 1.59-, 1.73-, and 1.76-fold, respectively.

For the composition of gut microbiota, Shannon, ACE, and Chao1 indexes were used to estimate species abundance and microbial diversity. A significant decrease in those indexes was found in the HCD group (Figure 6D–F, *p* < 0.0001) and *L. plantarum* WLPL21 had a reverse effect on that decrease (Figure 6D–F, *p* < 0.05).

From the view of bacterial diversity, there was a great change in the composition of gut microbiota after the administration of probiotics. At the phylum level, two main microbes were *Bacteroidetes* and *Firmicutes* in all groups; the *Firmicutes*-to-*Bacteroidetes* ratio (*F*/*B*) was 98.56% and 139.36%, respectively, in ND and HCD groups; by supplementation of *L. plantarum* WLPL21, that value was 70.87%. The *F*/*B* ratio was upregulated significantly in the HCD group as compared to the ND group (Figure 7C, *p* < 0.05), while treatment with *L. plantarum* WLPL21 significantly reversed this increase (Figure 7C, *p* < 0.01).

At the genus level, *Lactobacillus*, *Allbaculum*, and *Blautia* presented lower levels, while *Desulfovibro* and *Lachnoclostridium* had higher levels in the HCD group as compared with the ND group (Figure 7D, *p* < 0.05). The administration of *L. plantarum* WLPL21 significantly reversed that tendency as compared with the HCD group (Figure 7D, *p* < 0.05). To further disclose the relationship between gut microbiota and metabolite, a heatmap was performed and the results are shown in Figure 8, *Desulfovibrio* and *Lachnoclostridium* had a positive correlation with serum and liver cholesterol, while *Blautia*, *Allobaculum*, and *Lactobacillus* had a reverse effect. Moreover, *Lactobacillus* was found to have a positive correlation with cecum content SCFA level.

The LDA Effect Size (LEfSe) was constructed to further identify the specific bacterial taxa in each group. Our result showed that *Lactobacillus* was enriched in the WLPL21 group, and *Proteobacteria* was enriched in the HCD group (Figure 9A,B).

## 4. Discussion

Prevention of cardiovascular diseases is a great challenge for global research. Since a high-cholesterol diet probably causes hypercholesterolemia and led to cardiovascular diseases, in this study, three probiotic strains (*L. plantarum* WLPL21, WLPL72, and ZDY04) and two combinations (*L. plantarum* WLPL21 and *E. faecium* WEFA23, *L. plantarum* WLPL72 and *E. faecium* WEFA23) were approached for their alleviation effect on high-cholesterol-diet-induced hypercholesterolemia in mice. We found that *L. plantarum* WLPL21 was the best to alleviate high-cholesterol-diet-induced hypercholesterolemia by regulating cholesterol metabolism and transportation, and gut microbiota.

A reduction in cholesterol levels in the serum and liver reflected the capacity of probiotics for preventing hypercholesterolemia. Based on the comparison of TC, TG, HDL-C, and LDL-C in the liver and serum, as well as the TBA content in feces of seven test groups, only *L. plantarum* WLPL21 achieved ideal effects on hypercholesterolemia. Since the hepatic injury was closely related to the development of hypercholesterolemia, furthermore, *L. plantarum* WLPL21 also showed a critical role in hepatic protection.

Genetic polymorphisms were associated with chronic diseases. Zhang et al. reported that CYP17A1 gene polymorphisms were associated with lower serum total cholesterol levels in Han Chinese [26]. Chen et al. reported that the D19H polymorphism of ABCG8 could be considered a susceptible gene marker, indicating an increased likelihood of developing high cholesterol and LDL-C levels in Taiwanese consuming an ordinary Chinese diet [27]. Therefore, cholesterol metabolism and its transporters were important enzymes against hypercholesterolemia. Few publications reported that *Lactobacillus* spp. regulated cholesterol decomposition in the serum and liver and its transportation in the liver and ileum, thus alleviating hypercholesterolemia in mice. Yang et al. demonstrated that both *L. paracasei* 132 and *E. faecium* 201 alleviated hypercholesterolemia by regulating the expression of *Cyp7a1* and *Cyp8b1* in the liver to accelerate cholesterol decomposition [18]. Wan et al. declared that *L. plantarum* LRCC5273 ameliorated hypercholesterolemia by enhancing fecal cholesterol and bile acids excretion via increasing *Lxr* expression in the small intestine [28]. In our study, considering that hypercholesterolemia was mainly correlated with cholesterol decomposition and transportation, a systematic approach to the above relevant gene expression was focused, and we thus found that *L. plantarum* WLPL21 significantly upregulated the expression of decomposition genes (*Cyp7a1*, *Cyp7b1*, *Cyp27a1*, and *Cyp8b1*) (*p* < 0.05) in the liver, significantly upregulated that of transportation genes (*Abca1*, *Abcg5*, and *Abcg8* in liver, *Abcg5* and *Abcg8* in ileum) (*p* < 0.05), or downregulated *Npc1l1* expression in the ileum (*p* < 0.001). Moreover, cholesterol regulator genes (*Fxr*, *Lxr*, *Shp*, *Srebp*, and *Ppar-α*) (*p* < 0.05) had corresponding changes after the administration of *L. plantarum* WLPL21.

Regarding the fact that hypercholesterolemia might result in hepatic injury, subsequently, we monitored liver index and ALT and AST levels in the serum and hepatic H & E staining. We found that *L. plantarum* WLPL21 decreased the liver index and levels of ALT and AST, and prevented the liver from inflammation cell infiltration and lipid droplets steatosis. Similarly, *L. plantarum* H6 and T19 were proven to decrease ALT and AST levels in the serum [29], and *L. rhamnous* FJSY4-1 and *L. reuteri* FGSZY33L6 significantly degenerated hepatic cell vacuoles and reduced the area of lipid droplets [30].

Accompanied by hypercholesterolemia, as well as hepatic injury, the change in gut flora and its SCFAs metabolism was focused on by plenty of publications. In fact, for SCFAs, the intervention of hypercholesterolemia by lactic acid bacteria was positively related to its amount, as supported by our data and a few publications. However, the high agreement of gut flora, especially specific flora for the improvement of hypercholesterolemia, was less disclosed. We found that *L. plantarum* WLPL21 adjusted the *F*/*B* ratio in phylum level, significantly increased the abundance of *Lactobacillus*, *Allobaculum*, and *Blautia*, while it decreased the abundance of *Desulfovibro* and *Lachnoclostridium.* To our limited knowledge, the increase in *Allobaculum* and *Blautia* and the decrease in *Desulfovibro* and *Lachnoclostridium* resulting from the intervention of LAB have yet to be reported. Jing et al. reported that the abundance of *Allobaculum* and *Blautia* was increased after the supplementation of *Bacillus sp.* DU-106 and metformin [31]. Indeed, LAB intervention for hypercholesterolemia resulted in the change in different species. For example, Yang et al. showed that *L. fermentum* ZJUIDS06 and *L. plantarum* ZY08 improved the relative abundance of *Parabacteroides*, whereas *L. plantarum* H6 increased that of *Muribaculaceae* [32,33].

In previous publications, the cholesterol-lowering effect was closely related to abundance in genus level in the complex of cecum; however, the genus or even species in gut flora was less reported for the identification of its direct relationship with hypercholesterolemia. In our results, the abundance of *Allobaculum* and *Blautia* was negatively correlated with TC and LDL-C, respectively, in the serum and liver. Similarly, the supplementation of *L. plantarum* HT121 upregulated the level of *Blautia* to decrease serum lipid, indicating that *Allobaculum* and *Blautia* were probably bioinoculants to decrease TC and LDL-C, respectively, in the serum and liver [21]. Plenty of publications demonstrated that the abundance of *Lactobacillus* was positively correlated with the production of SCFAs, especially butyrate [34,35]. Moreover, *Lachnoclostridium* and *Desulfovibrio* might be responsible for an increase in TC and LDL-C, respectively, in the serum and liver. *Desulfovibrio* and *Lachnoclostridium* were significantly more abundant in the MC group and decreased after the supplementation of *L. plantarum* Y15 [36], indicating that *L. plantarum* WLPL21 alleviated hypercholesterolemia by increasing the abundance of *Allobaculum*, *Blautia*, and *Lactobacillus* and reducing that of *Lachnoclostridium* and *Desulfovibrio*.

In our study, only C57BL/6 male mice were involved for the treatment of *L. plantarum* WLPL21, considering its feasibility for forming high-cholesterol-diet-induced hypercholesterolemia. For expansion of our perspective results, other animal models, e.g., hypercholesterolemia rats, or even human clinical tests of hypercholesterolemia should be performed by using *L. plantarum* WLPL21 as an intervention strain.

## 5. Conclusions

Probiotics, especially *L. plantarum* WLPL21, reduced lipids in the serum and liver of mice, and modulated gut microbiota to alleviate hypercholesterolemia via detecting the concentration of TC and LDL-C, as well as SCFAs; the transcription of cholesterol metabolism and transportation; and the constitution of gut microbiota. Until now, the intervention of chronic diseases (e.g., hypercholesterolemia, type 2 diabetes, atherosclerosis) by probiotics and bioinoculants was prospective for improving human health. Besides *Lactobacillus* spp., *Allobaculum* and *Blautia* should be useful for the prevention and cure of hypercholesterolemia. Interestingly, in this approach, a combination of *E. faecium* WEFA23 with *L. plantarum* WLPL21 was found not to achieve a better effect than *L. plantarum* WLPL21 alone. Therefore, future exploration of the alleviation of hypercholesterolemia will be performed by fecal microbiota transplantation with *Allobaculum* and *Blautia*.

## Figures and Tables

**Figure 1 nutrients-15-02600-f001:**
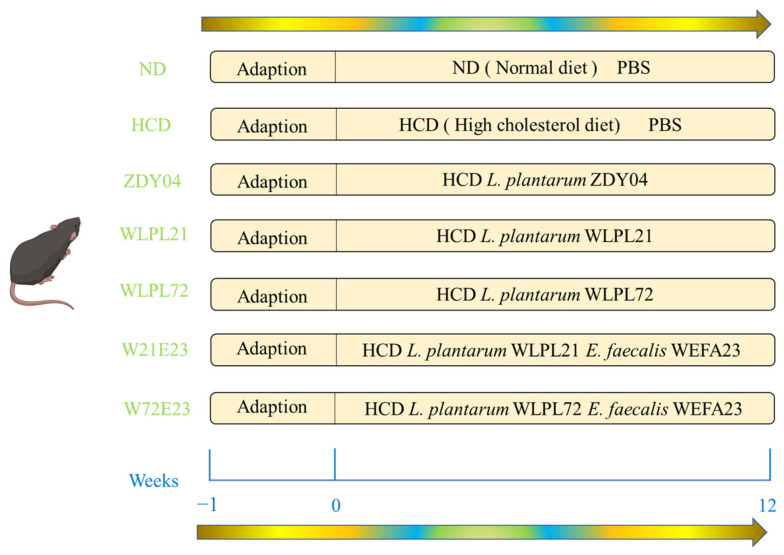
Diagram of animal experiment scheme.

**Figure 2 nutrients-15-02600-f002:**
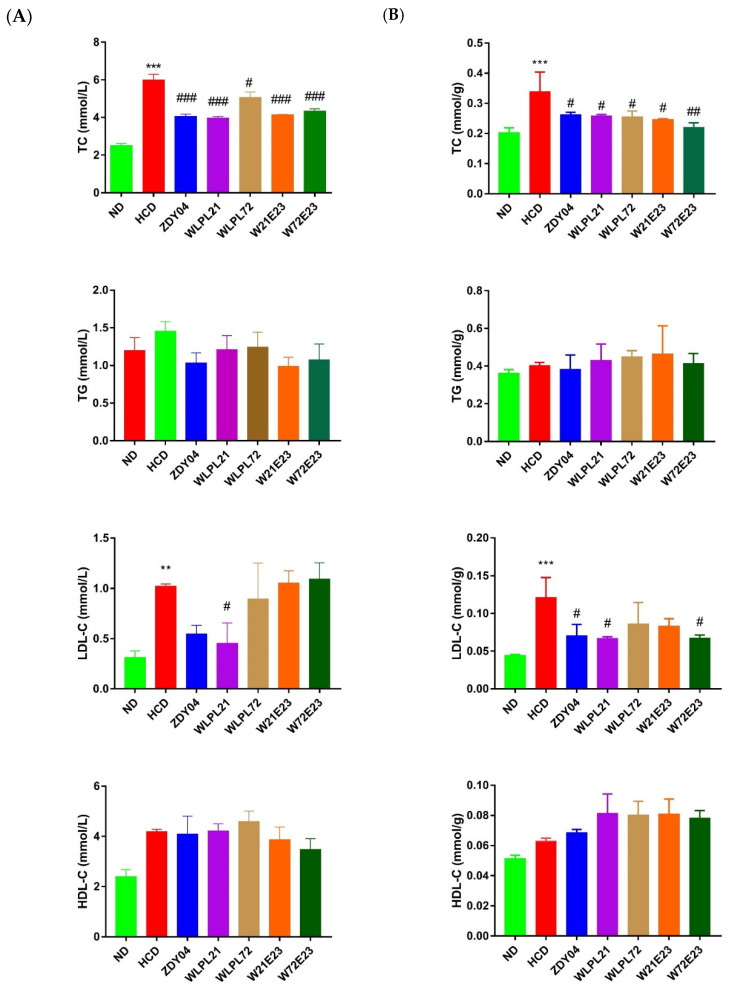
Effect of probiotics on the decrease in lipids level in serum (**A**) and liver (**B**) of mice. Total cholesterol (TC), triglycerides (TG), high-density-lipoprotein cholesterol (HDL-C), low-density-lipoprotein cholesterol (LDL-C). ND: normal diet; HCD: high-cholesterol diet; ZDY04: HCD with *Lactiplantibacillus plantarum* ZDY04; WLPL21: *L. plantarum* WLPL21; WLPL72: *L. plantarum* WLPL72; W21E23: HCD with bacterial cocktail of *L. plantarum* WLPL21 and *Enterococcus faecium* WEFA23; W72E23: HCD with bacterial cocktail of *L. plantarum* WLPL21 and *E. faecium* WEFA23. Data were expressed as mean ± SD. ** *p* < 0.01, *** *p* < 0.001, data in HCD were compared to ND; ^#^ *p* < 0.05, ^##^ *p* < 0.01, ^###^ *p* < 0.001, data in probiotic groups were compared to HCD.

**Figure 3 nutrients-15-02600-f003:**
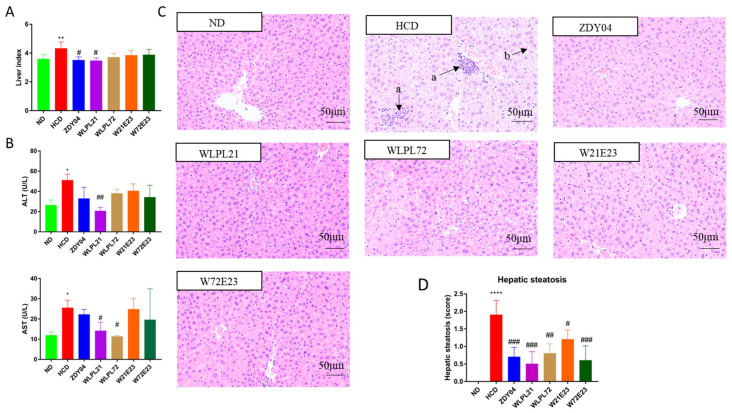
Effect of probiotics on hepatic injury caused by HCD. Liver index (**A**), serum aminotransferase of alanine (ALT) and aspartate (AST) (**B**), H & E staining of the liver (×200) (**C**) (a, inflammatory cell infiltration; b, lipid droplets accumulation), and hepatic steatosis scores of hypercholesterolemic mice livers (n = 5 per group) (**D**). ND: normal diet; HCD: high-cholesterol diet; ZDY04: HCD with *L. plantarum* ZDY04; WLPL21: *L. plantarum* WLPL21; WLPL72: *L. plantarum* WLPL72; W21E23: HCD with bacterial cocktail of *L. plantarum* WLPL21 and *E. faecium* WEFA23; W72E23: HCD with bacterial cocktail of *L. plantarum* WLPL21 and *E. faecium* WEFA23. Data were expressed as mean ± SD. * *p* < 0.05, ** *p* < 0.01, **** *p* < 0.0001, data in HCD were compared to ND; ^#^
*p* < 0.05, ^##^
*p* < 0.01, ^###^
*p* < 0.001, data in probiotic groups were compared to HCD.

**Figure 4 nutrients-15-02600-f004:**
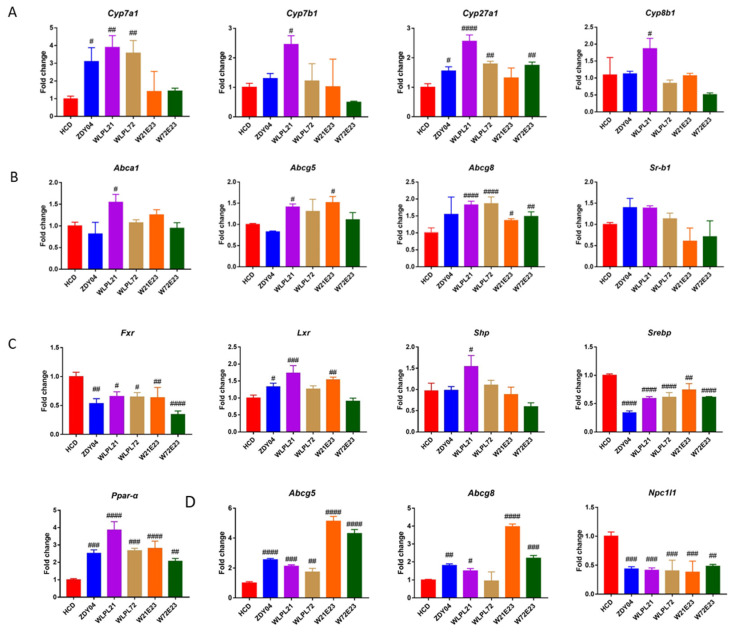
Effect of probiotics on expression of gene related to cholesterol metabolism (**A**), transport (**B**), and transcription regulatory factor (**C**) in the liver, and cholesterol transport in ileum (**D**). ND: normal diet; HCD: high-cholesterol diet; ZDY04: HCD with *L. plantarum* ZDY04; WLPL21: *L. plantarum* WLPL21; WLPL72: *L. plantarum* WLPL72; W21E23: HCD with bacterial cocktail of *L. plantarum* WLPL21 and *E. faecium* WEFA23; W72E23: HCD with bacterial cocktail of *L. plantarum* WLPL21 and *E. faecium* WEFA23. Data were expressed as mean ± SD. ^#^
*p* < 0.05, ^##^
*p* < 0.01, ^###^
*p* < 0.001, ^####^
*p* < 0.0001; data in probiotic groups were compared to HCD.

**Figure 5 nutrients-15-02600-f005:**
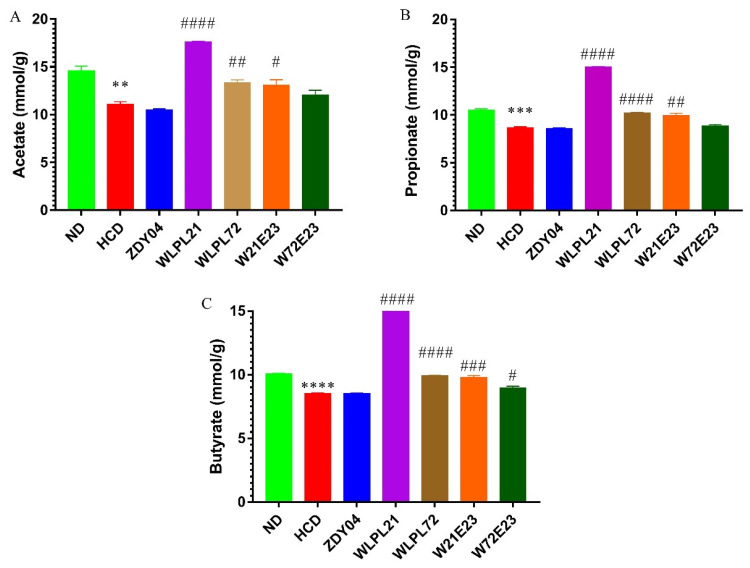
Effect of probiotics on level of acetate (**A**), propionate (**B**), and butyrate (**C**) in cecal content. ND: normal diet; HCD: high-cholesterol diet; ZDY04: HCD with *L. plantarum* ZDY04; WLPL21: *L. plantarum* WLPL21; WLPL72: *L. plantarum* WLPL72; W21E23: HCD with bacterial cocktail of *L. plantarum* WLPL21 and *E. faecium* WEFA23; W72E23: HCD with bacterial cocktail of *L. plantarum* WLPL21 and *E. faecium* WEFA23. Data were expressed as mean ± SD. ** *p* < 0.01, *** *p* < 0.001, **** *p* < 0.0001, data in HCD were compared to ND; ^#^
*p* < 0.05, ^##^
*p* < 0.01, ^###^
*p* < 0.001, ^####^
*p* < 0.0001, data in probiotic groups were compared to HCD.

**Figure 6 nutrients-15-02600-f006:**
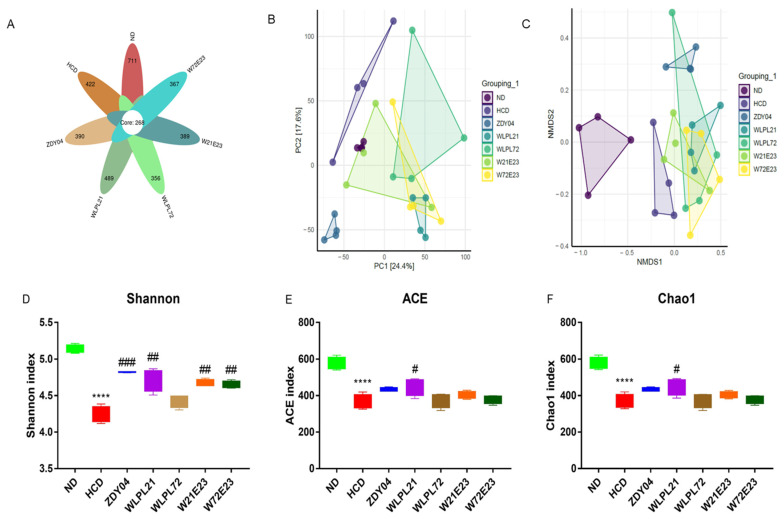
Effect of probiotics on species diversity abundance. Unique flora (**A**), principal component analysis (PCA) (**B**), non-metric multidimensional scaling (NMDS) (**C**), and Shannon (**D**), ACE (**E**), and Chao1 indexes (**F**) are shown. ND: normal diet; HCD: high-cholesterol diet; ZDY04: HCD with *L. plantarum* ZDY04; WLPL21: *L. plantarum* WLPL21; WLPL72: *L. plantarum* WLPL72; W21E23: HCD with bacterial cocktail of *L. plantarum* WLPL21 and *E. faecium* WEFA23; W72E23: HCD with bacterial cocktail of *L. plantarum* WLPL21 and *E. faecium* WEFA23. Data were expressed as mean ± SD. **** *p* < 0.0001, data in HCD were compared to ND; ^#^ *p* < 0.05, ^##^ *p* < 0.01, ^###^ *p* < 0.001, data in probiotic groups were compared to HCD.

**Figure 7 nutrients-15-02600-f007:**
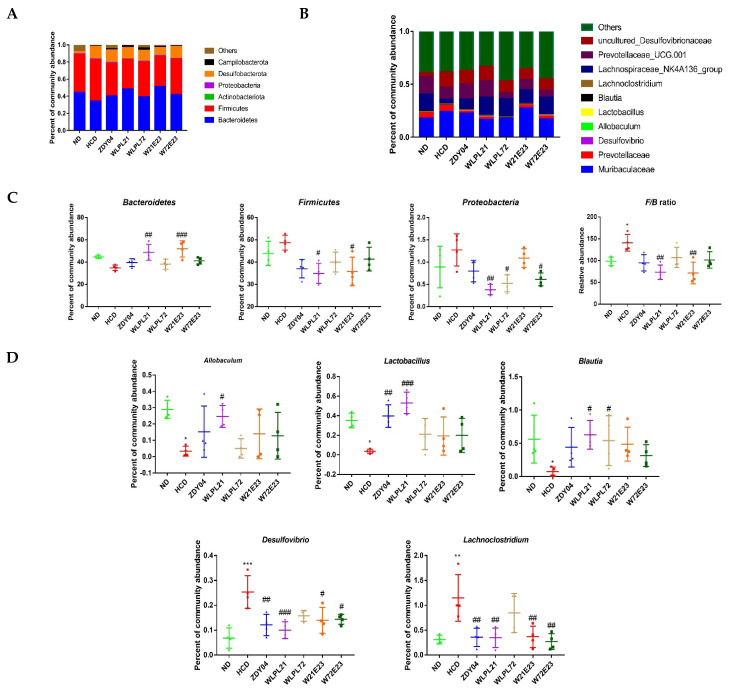
Effect of probiotics on gut microbial community composition in level of phylum (**A**) and genus (**B**), and type of high relative abundance of differential bacteria in level of phylum (**C**) and genus (**D**). ND: normal diet; HCD: high-cholesterol diet; ZDY04: HCD with *L. plantarum* ZDY04; WLPL21: *L. plantarum* WLPL21; WLPL72: *L. plantarum* WLPL72; W21E23: HCD with bacterial cocktail of *L. plantarum* WLPL21 and *E. faecium* WEFA23; W72E23: HCD with bacterial cocktail of *L. plantarum* WLPL21 and *E. faecium* WEFA23. Data were expressed as mean ± SD. * *p* < 0.05, ** *p* < 0.01, *** *p* < 0.001, data in HCD were compared to ND, ^#^ *p* < 0.05, ^##^ *p* < 0.01, ^###^ *p* < 0.001, data in probiotic groups were compared to HCD.

**Figure 8 nutrients-15-02600-f008:**
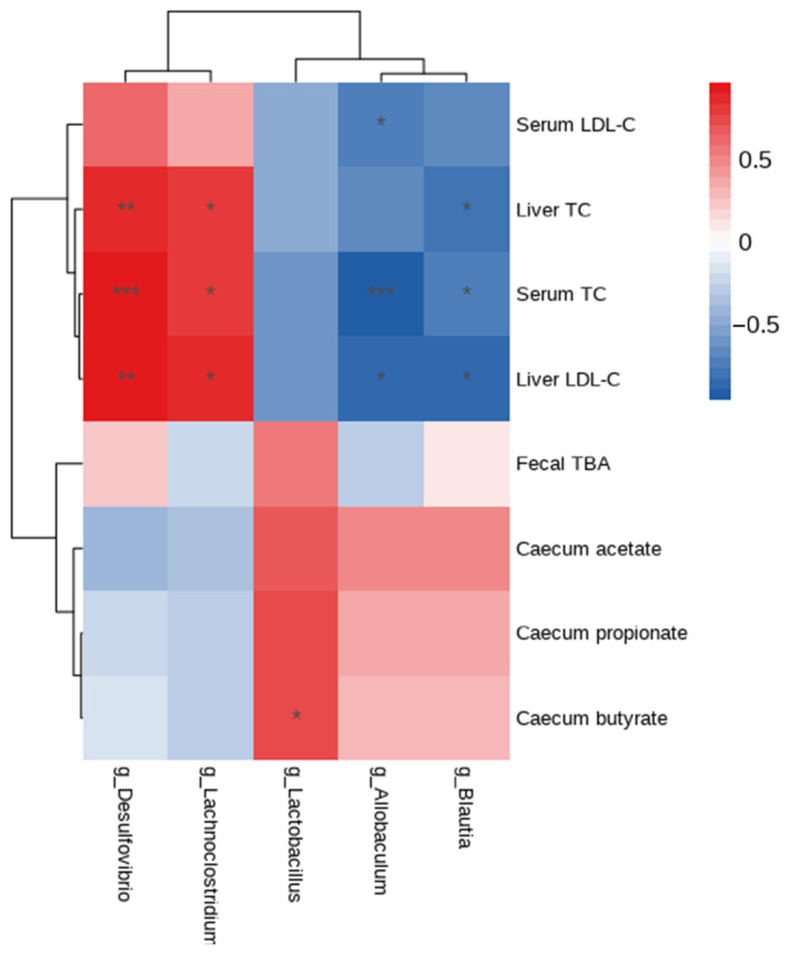
Correlation heatmap between bacteria at genus level and major metabolite including serum and liver lipids, SCFAs, and fecal TBA. * *p* < 0.05, ** *p* < 0.01, *** *p* < 0.001.

**Figure 9 nutrients-15-02600-f009:**
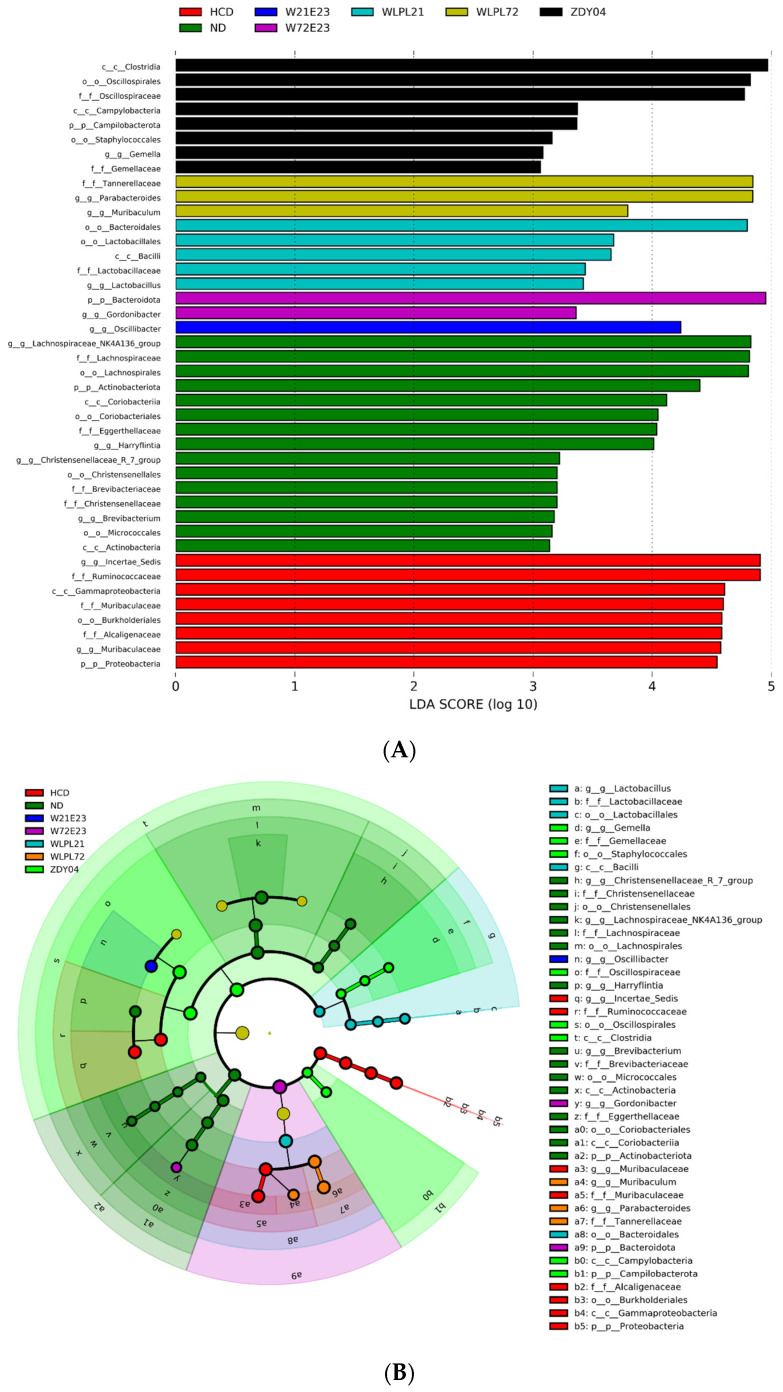
Identification of most characteristic taxa by linear discriminant analysis (LDA) effect size (LEfSe). The most significant difference of gut microbial taxa among groups after LDA (**A**). Cladogram visualizing the output of the LEfSe analysis (**B**). The threshold on the logarithmic LDA score for discriminative features was set to 3.0. The bar length of LDA represents the influence of species abundance on the difference effect. p, phylum; c, class; o, order; f, family; and g, genus. ND: normal diet; HCD: high-cholesterol diet; ZDY04: HCD with *L. plantarum* ZDY04; WLPL21: *L. plantarum* WLPL21; WLPL72: *L. plantarum* WLPL72; W21E23: HCD with bacterial cocktail of *L. plantarum* WLPL21 and *E. faecium* WEFA23; W72E23: HCD with bacterial cocktail of *L. plantarum* WLPL21 and *E. faecium* WEFA23.

**Table 1 nutrients-15-02600-t001:** The sequence of primers used for RT-qPCR.

Gene	Forward Primer (5′→3′)	Reverse Primer (5′→3′)
*Gapdh*	AGGTCGGTGTGAACGGATTTG	GGGGTCGTTGATGGCAACA
*Cyp27a1*	GCCTTGGAAGCCATCACCTA	AGATCTGATGAAGGCGGCAG
*Cyp7b1*	CCCTGCGTGACGAAATTGAC	AGAATAGTGCTTTCCAGGCAGA
*Cyp7a1*	AGCAACTAAACAACCTGCCAGTA	GTCCGGATATTCAAGGATGCA
*Cyp8b1*	TTGCAAATGCTGCCTCAACC	AGTGGGAAATTAACAGTCGCA
*Abca1*	GTTGGTCTCCAGAAGGTATT	TTCAGGATGTCCATGTTGT
*Abcg5*	TCAATGAGTTTTACGGCCTGAA	GCACATCGGGTGATTTAGCA
*Abcg8*	TGCCCACCTTCCACATGTC	ATGAAGCCGGCAGTAAGGTAGA
*Sr-b1*	TGTACTGCCTAACATCTTGGTCC	ACTGTGCGGTTCATAAAAGCA
*Fxr*	TGAGAACCCACAGCATTTCG	GCGTGGTGATGGTTGAATGTC
*Lxr*	CTCAATGCCTGATGTTTCTCCT	TCCAACCCTATCCCTAAAGCAA
*Shp*	CGATCCTCTTCAACCCAGATG	AGGGCTCCAAGACTTCACACA
*Srebp1c*	GGAGCCATGGATTGCACATT	GGCCCGGGAAGTCACTGT
*Ppar-α*	AACATCGAGTGTCGAATATGTGG	CCGAATAGTTCGCCGAAAGAA
*Npc1l1*	TTTCTAGGGGCCCTGACCTC	TTGAAAAGCAGCACACGACG

## Data Availability

The data used to support the findings of this study are available from the corresponding author upon request.

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
