# Peer review of "Alleviation Syndrome of High-Cholesterol-Diet-Induced Hypercholesterolemia in Mice by Intervention with Lactiplantibacillus plantarum WLPL21 via Regulation of Cholesterol Metabolism and Transportation as Well as Gut Microbiota"

_nutrients, 2023, doi:10.3390/nu15112600_

Round 1

Reviewer 1 Report

The probiotics are of great prospects in respect of prevention and treatment of cardiovascular diseases, hence the paper relates a highly relevant and hot topic. My questions are as follows:

What other bacteria proved to be efficient with regard to amelioration of hypercholesterolemia? Why the applied LAB species were selected?

Why sys- 15 tematic study was selected for the tool to be applied in the course of the research?

It was established that a huge change occurs in the composition of gut microbiota after the administration of probiotics, however just a limited number of strains was involved. Could you explain this dichotomy?

Some grammatical eerors are to be perceived.

Reviewer 2 Report

In my opinion - a missing part at the end of the discussion is entitled 'Limitations of paper'. This chapter should have included aspects such as using animal model only and caution when interpreting studies on mice and drawing conclusions about clinical aspects. In addition, the results refer to analyses on males, so additional caution regarding the whole population (irrespective of sex).
It is important that an explanation of abbreviations is provided immediately - and not in the chapter entitled MATERIALS AND METHODS.
More information is needed on the statistical analyses used. Polymorphisms in enzymes important in cholesterol metabolism and its transporters should be highlighted in the discussion.
I do not agree with the sentence in the INTRODUCTION: “ Hypercholesterolemia, a leading cause of numerous chronic diseases e.g., cardiovascular diseases (CVDs) [1], hypertension [2], diabetes [3] and kidney failure [4], is recognized as a complex metabolic disorder owing to the accumulation of cholesterol in cells and plasma of blood [5]”, because hypercholesterolemia is not important in this regard, but dyslipidemia with low HDL cholesterol and high LDL cholesterol, especially in the form of oxidized, atherogenic form of LDL.
The work is based on current literature. Less than 12% (4 items out of 34) are older than 5 years. The choice of literature, and its collation - shows scientific maturity (a good part of the work).
Minor mistakes:
Instead:
It was observed that TC level in serum and liver was both significantly decreased in all probiotics treatment (p < 0.05), just similar as the above result of l body weight loss.
Should be:
It was observed that TC level in serum and liver was significantly decreased in all probiotics treatments (p < 0.05), similar to the above result of l body weight loss.

Instead:
Few publications reported that Lactobacillus spp regulated cholesterol decomposition in serum and liver, and its transportation in liver and ileum, and thus alleviated hypercholesterolemia in mice.
Should be:
Few publications reported that Lactobacillus spp. regulated cholesterol decomposition in serum and liver and its transportation in the liver and ileum, thus alleviating hypercholesterolemia in mice.

Overall, the study presented in the paper is interesting.

In my opinion - a missing part at the end of the discussion is entitled 'Limitations of paper'. This chapter should have included aspects such as using animal model only and caution when interpreting studies on mice and drawing conclusions about clinical aspects. In addition, the results refer to analyses on males, so additional caution regarding the whole population (irrespective of sex).
It is important that an explanation of abbreviations is provided immediately - and not in the chapter entitled MATERIALS AND METHODS.
More information is needed on the statistical analyses used. Polymorphisms in enzymes important in cholesterol metabolism and its transporters should be highlighted in the discussion.
I do not agree with the sentence in the INTRODUCTION: “ Hypercholesterolemia, a leading cause of numerous chronic diseases e.g., cardiovascular diseases (CVDs) [1], hypertension [2], diabetes [3] and kidney failure [4], is recognized as a complex metabolic disorder owing to the accumulation of cholesterol in cells and plasma of blood [5]”, because hypercholesterolemia is not important in this regard, but dyslipidemia with low HDL cholesterol and high LDL cholesterol, especially in the form of oxidized, atherogenic form of LDL.
The work is based on current literature. Less than 12% (4 items out of 34) are older than 5 years. The choice of literature, and its collation - shows scientific maturity (a good part of the work).
Minor mistakes:
Instead:
It was observed that TC level in serum and liver was both significantly decreased in all probiotics treatment (p < 0.05), just similar as the above result of l body weight loss.
Should be:
It was observed that TC level in serum and liver was significantly decreased in all probiotics treatments (p < 0.05), similar to the above result of l body weight loss.

Instead:
Few publications reported that Lactobacillus spp regulated cholesterol decomposition in serum and liver, and its transportation in liver and ileum, and thus alleviated hypercholesterolemia in mice.
Should be:
Few publications reported that Lactobacillus spp. regulated cholesterol decomposition in serum and liver and its transportation in the liver and ileum, thus alleviating hypercholesterolemia in mice.

Overall, the study presented in the paper is interesting.
